# *Angiostrongylus cantonensis* an Atypical Presenilin: Epitope Mapping, Characterization, and Development of an ELISA Peptide Assay for Specific Diagnostic of Angiostrongyliasis

**DOI:** 10.3390/membranes12020108

**Published:** 2022-01-19

**Authors:** Salvatore G. De-Simone, Paloma Napoleão-Pêgo, Priscila S. Gonçalves, Guilherme C. Lechuga, Arnaldo Mandonado, Carlos Graeff-Teixeira, David W. Provance

**Affiliations:** 1Center of Technological Development in Health (CDTS), National Institute of Science and Technology for Innovation on Neglected Diseases (INCT-IDN), FIOCRUZ, Rio de Janeiro 21040-900, RJ, Brazil; paloma.pego@cdts.fiocruz.br (P.N.-P.); priscilla.desimone@cdts.fiocruz.br (P.S.G.); guilherme.lechuga@cdts.fiocruz.br (G.C.L.); bill.provance@cdts.fiocruz.br (D.W.P.J.); 2Laboratory of Epidemiology and Molecular Systematics (LESM), Oswaldo Cruz Institute, FIOCRUZ, Rio de Janeiro 21040-900, RJ, Brazil; 3Department of Cellular and Molecular Biology, Biology Institute, Federal Fluminense University, Niterói 24220-900, RJ, Brazil; 4Laboratory of Biology and Parasitology of Wild Mammals Reservoirs, Oswaldo Cruz Institute, FIOCRUZ, Rio de Janeiro 21040-360, RJ, Brazil; maldonado@ioc.fiocruz.br; 5Infectious Diseases Unit, Department of Pathology, Federal University of Espirito Santo, Vitória 29075-910, ES, Brazil; carlos.teixeira@ufes.br

**Keywords:** *Angiostrongylus cantonensis*, transmembrane protease, aspartyl protease, presenilin-like, characterization, B epitopes, immunological diagnosis, ELISA

## Abstract

Background: Angiostrongyliasis, the leading cause universal of eosinophilic meningitis, is an emergent disease due to *Angiostrongylus cantonensis* (rat lungworm) larvae, transmitted accidentally to humans. The diagnosis of human angiostrongyliasis is based on epidemiologic characteristics, clinical symptoms, medical history, and laboratory findings, particularly hypereosinophilia in blood and cerebrospinal fluid. Thus, the diagnosis is difficult and often confused with those produced by other parasitic diseases. Therefore, the development of a fast and specific diagnostic test for angiostrongyliasis is a challenge mainly due to the lack of specificity of the described tests, and therefore, the characterization of a new target is required. Material and Methods: Using bioinformatics tools, the putative presenilin (PS) protein C7BVX5-1 was characterized structurally and phylogenetically. A peptide microarray approach was employed to identify single and specific epitopes, and tetrameric epitope peptides were synthesized to evaluate their performance in an ELISA-peptide assay. Results: The data showed that the *A. cantonensis* PS protein presents nine transmembrane domains, the catalytic aspartyl domain [(XD (aa 241) and GLGD (aa 332–335)], between TM6 and TM7 and the absence of the PALP and other characteristics domains of the class A22 and homologous presenilin (PSH). These individualities make it an atypical sub-branch of the PS family, located in a separate subgroup along with the enzyme *Haemogonchus contournus* and separated from other worm subclasses. Twelve B-linear epitopes were identified by microarray of peptides and validated by ELISA using infected rat sera. In addition, their diagnostic performance was demonstrated by an ELISA-MAP4 peptide. Conclusions: Our data show that the putative AgPS is an atypical multi-pass transmembrane protein and indicate that the protein is an excellent immunological target with two (PsAg3 and PsAg9) *A. costarisencis* cross-reactive epitopes and eight (PsAg1, PsAg2, PsAg6, PsAg7, PsAg8, PsAg10, PsAg11, PsAg12) apparent unique *A. cantonensis* epitopes. These epitopes could be used in engineered receptacle proteins to develop a specific immunological diagnostic assay for angiostrongyliasis caused by *A. cantonensis*.

## 1. Introduction

Angiostrongyliasis is a parasitic zoonosis disease caused by the nematode species *Angiostrongylus cantonensis* and *A. costaricensis*. It is endemic in the Southeast Asia-Pacific Basin region and the American continent (the Caribbean and South America) [1,2,3]. Still, the parasite has also been found in Australia, some areas of Africa, Hawaii, Louisiana, and more recently in Europe [4]; it is reported that at least 2800 cases have been recorded globally [5]. *A. costaricensis* causes an abdominal disease, while the *A. cantonensis* an eosinophilic meningoencephalitis. The first was described in Costa Rica in 1952 [6,7,8] and the second infecting pulmonary arteries and hearts of domestic rats in Guangzhou (Canton), China, by Chen in 1935 [9]. In both cases, rats are the primary hosts, and snails are the intermediate hosts [10,11].

Diagnosis is sought via clinical criteria, including the presence of cerebrospinal fluid eosinophils and a history of exposure to *A. cantonensis* larvae (third-larvae, L3), e.g., from raw freshwater snails (Pila or Pomacea snails) or contaminated vegetables [12,13]. Conversely, the diagnosis based on clinical presentation is difficult because the symptoms are often confused with those produced by other parasitic diseases such as paragonimiasis, gnathostomiasis, and cysticercosis [14]. A definitive diagnosis is confirmed by discovering the worms in cerebrospinal fluid after a lumbar puncture or in the eyes during surgery, but direct detection of the parasite in patients is rare [2,15].

Over the past decades, many non-particulate proteins (adult worms, brain-stage larvae, or excretory-secretory products) have been evaluated as potential targets for the development of immunology to support the clinical diagnosis [16,17,18,19]. The enzyme-linked immunosorbent assay (ELISA) [20,21,22,23], dot blot [16,23,24,25], and immunoblot [26] with extract of the parasite [27,28] or using proteins isolated by electroelution [27] have been most widely applied. More recently, recombinant proteins derived from these previously identified proteins were also applied to a lateral-flow immunochromato graphic test assay [29,30,31]. However, in the totality of these works, antigens highly conserved in nematodes and other correlated genes were used, which may suggest cross-reactive epitopes [32,33], and their specificity and validation have not yet been present bee confirmed in studies involving multi diseases patients sera.

Next-generation metagenomic sequencing for diagnosing angiostrongyliasis of the central nervous system is being validated as a possible diagnostic methodology [34]. However, the fact that the number of *A. cantonensis* DNA readings has fluctuated with the opening pressure of the CSF and clinical manifestations calls into question the validity of the methodology for laboratory diagnosis [35].

Therefore, given the difficulties of applying molecular tools, the development of immunological methods continues to be a viable and necessary alternative for the quick and accurate diagnosis of angiostrongyliasis. Consequently, initial protein characterization is crucial in reaching an excellent immunological diagnostic or therapeutic target [25,36].

Proteases are critical elements of several biological processes in living cells and, therefore, represent one of the most prominent families of biopharmaceutical targets. Among this group of proteins, the traditional family of aspartyl proteases stands out, and within these, a new group of polytopic membrane proteins called presenilin (PS) presents extremely divergent sequences outside of some well-conserved motifs. This group of proteins has hardly been studied for drug development [37] but may also serve as antigens [38,39,40], and the discovery of new families of membrane-associated aspartyl proteases is growing in eukaryotic cells [41,42,43]. Notably, it has been estimated that proteases account for 5 to 10% of all drug targets studied for therapeutic development [10].

Conversely, to inhibit and block the activity of pathogenic proteases or to identify specific epitopes with the desired selectivity, monoclonal antibodies (mAbs) have shown great promise as research tools, therapeutic agents, and diagnostics. However, identifying mAbs with inhibitory functions is challenging because current antibody discovery methods rely on binding rather than inhibition. Consequently, identifying epitopes located in strategic domains of these proteins through peptide microarray presents a faster and more efficient alternative to identify these determinants and then produce the desired antibodies.

Therefore, in this study, we describe the partial biochemical and immunological characterization of a putative presenilin protein of *A. cantonensis* and its epitopes in developing an ELISA test with high specificity and sensitivity for the diagnosis of the angiostrongyliasis.

## 2. Materials and Methods

### 2.1. Rat Infected Sera and Ethics Statement

Sera samples from 16 infected rats in the acute phase of angiostrongyliasis were obtained from the Laboratory of Biology and Parasitology of Wild Mammals Reservoirs (Oswaldo Cruz Institute, FIOCRUZ, Brazil). Another group included 25 uninfected rat sera.

### 2.2. Synthesis of the Cellulose-Membrane-Bound Peptide Array

A library of eight one, 15-mer peptides were designed to represent a consecutive overlapping coverage offset by nine amino acids across the entire coding region (415 aa) of the putative aspartyl protease (C7BVX5_ANGCA, UNIPROT) of *A. cantonensis.* The peptides were automatically prepared onto amino-PEG_500_-UC540 cellulose membranes (Intavis Bioanalytical Instruments, Köln, Germany) as previously described [44] using an Auto-Spot Robot ASP-222 (Intavis Bioanalytical Instruments AG, Köln, Germany) and the F-moc strategy. In brief, coupling reactions were followed by acetylation with acetic anhydride (4%, *v*/*v*) in *N*,*N*-dimethylformamide to render peptides unreactive during the subsequent steps. After acetylation, F-moc protective groups were removed by adding piperidine to render nascent peptides reactive. The remaining amino acids were added by this same process of coupling, blocking, and deprotection until the expected desired peptide was generated. After adding the last amino acid in the peptide, the amino acid side chains were deprotected using a solution of dichloromethane–trifluoracetic acid–tri isobutyl silane (1:1:0.05, *v*/*v*/*v*) and washed with methanol. Membranes containing the synthetic peptides were either probed immediately or stored at −20 °C until required. Negative controls [without peptide; IHLVNNESSEVIV HK (*Clostridium tetani*) precursor peptide] and positive controls were included in each assay.

### 2.3. Screening of SPOT Membranes

SPOT membranes were washed with TBS (50 mM Tris-buffer saline, pH 7.0) and blocked with TBS-CT (Tris-buffer saline, 3% casein, 0.1% Tween 20, pH 7.0) at room temperature under agitation or overnight at 4 °C. After extensive washing with TBS-T (Tris-buffer saline, 0.1% Tween 20, pH 7.0), the membrane presenting the peptide library was incubated for two hours with a pool of patients sera (1:100) in TBS-CT and then washed again with TBS-T. Afterward, the membrane was incubated with alkaline phosphatase-labeled goat anti-rat IgG (1:5000 in TBS-CT; KPL, Gaithersburg, MD, USA) for one hour, and then washed with TBS-T and CBS (50 mM citrate-buffer saline, pH 7.0). Chemiluminenscente CDP-Star^®^ Substrate (0.25 mM) with Nitro-Block-II™ Enhancer (Applied Biosystems, Waltham, MA, USA) was added to complete the reaction.

### 2.4. Scanning and Measurement of Spot Signal Intensities

As described previously, chemiluminescent signals were detected on an Odyssey FC (LI-COR Bioscience, Lincoln, NE, USA) [45]. Briefly, a digital image file was generated at a resolution of 5 MP, and the signal intensities were quantified using the TotalLab TL100 (v. 2009, Nonlinear Dynamics, Newcastle, Tyne, UK) software. The signal intensity (SI) used as a background was a set of negative controls spotted in each membrane.

### 2.5. Synthesis of Peptides and Preparation of the MAPs

Nine single *A. cantonensis* peptides (PSAg1-PSAg3, PSAg6-PSAg9, PSAg11, and PSAg12; Table 1) were synthesized by the F-moc strategy in a synthesizer machine (MultiPep-1 CEM, Corp, Charlotte, NC, USA). For the preparation of the dendrimer multi-antigen peptides (MAP4), the same synthesis protocol was used in the solid phase and the tetrameric F-moc-Lys2-Lys-β-Ala Wang resin, as described previously [46]. The constructs were prepared in the automated MultiPep-1 peptide synthesizer, and the side chains of tetrafunctional F-moc-amino acids were protected with TFA-labile protecting groups as required. Residues corresponding to the monovalent (‘tail’) part of the construct, up to the first (bis-Fmoc) Lys residue initiating the dendrimer structure, were incorporated via single couplings. Once sequence assembly was completed, the F-moc groups were removed, and the peptide-resin was cleaved and fully deprotected with TFA/H2O/EDT/TIS (94/2.5/2.5/1.0 *v*/*v*, 90 min). The peptide was precipitated by adding chilled diethyl ether, centrifuged for 3 × 10 min at 4 °C, and the pellet was taken up in aqueous AcOH (10% *v*/*v*), dried, and stored as a lyophilized powder. When necessary, the MAP was dissolved in water, centrifuged (10,000 g, 60 min, 15 °C) and the supernatant filtered on a centricon 10 filter. The single peptides were used without previous purification, but their identity was checked by MS (MALDI-TOF or electrospray).

### 2.6. ELISA-Peptide

The ELISA was realized, as described previously [47], with minor modifications. Briefly, ELISA plates (Immulon 4HB; Corning, NY, USA) were coated with 80 µL (50 µg) of each peptide prepared on coating buffer (Na_2_CO_3_–NaHCO_3_, pH 9.6) overnight at 4 °C. After each incubation step, the plates were washed three times using a PBS-T washing buffer (PBS with 0.1% Tween 20 adjusted to pH 7.2) and blocked (200 µL) with 2.5% BSA and incubated 2 h at 37 °C. The rat’s sera were diluted (1:150) in coating buffer, and 100 µL were applied onto immunosorbent plates and incubated for 2 h at 37 °C. Following several washes with PBS-T, the plates were incubated with 100 µL goat anti-rat IgG conjugated to HRP (1:5000 dilutions at blocking buffer; KPL, Gaithersburg, MD, USA) for 2 h. The plates were developed with p-nitrophenyl phosphate (p-NPP) as a substrate (Sigma-Aldrich, St. Louis, MO, USA). The absorbance values at 405 nm were read using a FlexStation 3 Microplate Reader (Molecular Devices, Sunnyvale, CA, USA), and the reactivity index was defined as O.D._450_ value of Target—O.D._450_ values of cut-off.

### 2.7. Database Searches, Computational, and Phylogeny Studies

To curate a family of 10 target sequences for C7BVX5-1, a set of aspartic protease sequences were retrieved from the Uniprot database (available online: http://www.uniprot.org/ accessed on 20 March 2021) using the following criteria: An EC designation of 3.4.23 for aspartic endopeptidases; a length between 300 to 800, and a reviewed annotation. Another set of sequences homologous to other proteins, which have been cited by Blast, was added after a delta-blast of the Refseq database with coverage greater than 80% of their sequence-issue and identity over 10–30%. From this set, an alignment was performed on the Mat server with blossum80 and a gap penalty of 2.5, followed by clustering to eliminate the excessive gaps with the method “minimum linkage.”

The potential transmembrane (TM) domains (TMD) of the *A. cantonensis* enzyme were analyzed by three different predictions programs; the TMpred [48], the TopCons [49], and MemConP [50]. The model most consistent with the identified epitopes and motifs were obtained using TopCons, which used an algorithm based on the statistical analysis of base. The prediction was made using a combination of several weight-matrices for scoring. Secondary structure predictions were obtained from PSIPRED (Available online: http://bioinf. cs.ucl.ac.uk/psipred/ accessed on 15 March 2021) and CDM (available online: http://gor.bb.iastate.edu/cdm/ accessed on 15 March 2021) servers. The tertiary structure prediction was performed on the LOMETS server (available online: http://zhanglab.ccmb.med.umich.edu/LOMETS/ accessed on 15 March 2021) and I-TASSER (Available online: https://zhanglab.ccmb.med.umich.edu/I-TASSER accessed on 15 November 2020). Finally, a network analysis was performed using STRING tutorial and the Cytoscape platform version 3.7.2; (available online: https://apps.cytoscape.org/apps/stringapp accessed on 21 October 2021) [51].

### 2.8. Ethics Statement

The study was approved by the FIOCRUZ (IOC-CAAE: 52892216.8.0000.5248 and 1.896.362) study center ethics committee and conducted following all applicable Good Clinical Practice (GCP) regulatory requirements, including the Declaration of Helsinki.

### 2.9. Statistical Analysis

Statistical analysis was performed using GraphPad Prism version 5.0. The statistical differences using a *t*-test were considered if the *p*-value ≤ 0.05.

## 3. Results

### 3.1. Identification and Mapping of Linear Epitopes Using Synthetic Peptides

The epitopes of the PS protein from *A. cantonensis* recognized by patients sera were mapped using the parallel Spot-synthesis strategy. The peptide library consisted of 69 peptide sequences of 15 amino acids that overlapped by 10 amino acids and covered the entire protein sequence. A representative experiment is presented in Figure 1. The list of synthesized peptides is shown in Appendix A. Overall, each epitope displayed a relatively strong reactivity (containing 4–15 residues of extension). However, the most vigorous intensity was observed with the antigenic determinant in the peptide SIFWKGPMRLQQASL (B10) and PSSGRFIESFRMPSL (C2).

The analysis of spot signal intensity for the synthesized peptides from the PS sequence showed 12 epitopes (Table 1). The secondary structure of the identified epitopes is also presented in Appendix A and was based on the results obtained by the I-Tasser prediction (accessed on 10 March 2021).

### 3.2. Absence of Crucial Motifs for A22 Aspartyl Protease Family

The data available from studies on the γ-secretase complexes in mammalian cells have indicated that particular amino acid motifs are crucial for its proteolytic activity, substrate recognition, and complex assembly [52]. To prove that these critical motifs are also conserved in the putative *A. cantonensis* protein available in the protein database UniProtKB (http://www.uniprot.org/; accessed on 3 March 2021) with the number C7BVX5 belonging to the A22 family, multiple sequence alignments were performed. Potential homologs were identified by PSI-BLAST, and the level of similarity among the PSs from these species was aligned to the *Homo sapiens* sequence. Motif analysis and multiple sequence alignment of the *A. cantonensis* PS sequence (C7BVX5) with the *H. sapiens* PS1 protein (P49768) was performed and showed only one compatible conserved motif sequence. Those of the catalytic pocket (YD; 192–194 and GLGD; 287–290) located in TM7 aligned with XD (aa 241) and GLGD (aa 332-335) of *A. cantonensis* PS, in TMD6 and TDM7, respectively (Figure 2).

Other motifs were also compared with PS proteins from other worms such as *Caenorhabditis elegans* (P52166), *Angiostrongylus costaricensis* (A0A0R3PEK8), and *Schistosoma mansoni* (C0SKM6). The human TMD8 has two motifs [GxxxG (GVKLG) and SxxxGxxxxA (SVLVGKASA)], which are localized close to the catalytic Asp. These motifs contain sites of FAD mutations, which form a portion of the catalytic core of the PSs and influence helix packing that may modulate enzymatic activity. In the *A. cantonensis* PS, these motifs were not identified. Another interesting difference is that, unlike humans and other nematodes, *A. cantonensis* PS lacks the conserved C-terminal PALP motif that determines the active site’s conformation and characterizes the A22 family (Figure 2).

### 3.3. Spatial Location of the Most Reactive Epitopes

To analyze the localization of the epitopes on the *A. cantonensis* protein, a 3D structure was obtained from the I-Tasser server (Figure 3A), and the position of the 12 epitopes (Table 1) was identified by SPOT-Synthesis was depicted. In addition, transmembrane domains, hydropathy, and secondary structures were analyzed to pinpoint the epitope position in the presenilin antigen (PsAg) protein.

Four identified epitopes were exclusively in loop/coil (PsAg1, 2, 8, and 12) structures, while five (PsAg3, 6, 7, 9, and 11) shared amino acids in coil and helix, sometimes in the interface between membrane and cytosol/extracellular space. Three identified peptides were found in the alpha-helix (PsAg4, 5, and 10) (Table 1, Figure 3A). Although the epitope PsAg10 was predicted to stay partially embedded in the membrane helix, probably due to an unspecific reaction, this peptide was excluded from subsequent analysis. However, all other epitopes were present on the protein surface or adjacent to the membrane and intracellular/extracellular space, accessible to the solvent. The hydropathy plots of the protein and the amino acid accessibility, shown in Appendix A, also suggested that residues in all of the epitopes were present on the surface of the protein.

Figure 3B show a transmembrane model determined for the AgPS using the TopCons software and the distribution of the identified epitopes along with the primary structure. The predicted model depicted nine TM domains, which is consistent with the classical structure of the PS proteins. In addition, the expected model contemplates the catalytic domain consisting of the conserved motifs XD and GLGD, between TM6 and TM7.

### 3.4. Phylogeny and Protein-Protein Interaction

A group of sequences for PSs was retrieved on the Uniprot server (Available online: http://www.uniprot.org accessed on 23 August 2020) through a search using the criteria name, “PS,” and a length of 300 to 800. A phylogenetic tree was created with the sequences aligned by the neighbor-joining algorithm using a CLUSTALW program (Figure 4). The *A. cantonensis* aspartyl protease was localized in a separate subgroup along with the *Haemonchus contortus* enzyme detached from the other worm’s subclasses of this protease family, such as *Fasciola hepatica* and *Caenorhabditis elegans*.

### 3.5. Selection of Putative Specific Epitopes

The first goal to eliminate cross-reactive epitopes was to search possible homology sequences of the epitopes in the Pir data bank. The peptide search database realized with the 12-epitope sequences returned, that 2 (PsAg3 and PsAg9) cross-react with *A.*
*costaricensis* while 8 (PsAg1, PsAg2, PsAg6, PsAg7, PsAg8, PsAg10, PsAg11, PsAg12) are unique (Table 1; Appendix A). PsAg4, PsAg5, and PsAg10 are defined in a helical well; they were not interpreted as B-linear epitopes.

Then, to confirm the immunogenicity of the epitopes, nine single peptides were synthesized (Table 1) and individually evaluated by ELISA assay using serum from infected rats. The results (media ± SD) are shown in Figure 5 and show that all the epitopes were highly reactive.

Thus, one antigenic determinant (PsAg2) was selected and synthesized as MAP4 containing 3 Gly in the N and 2 G in the C terminus (*GGG***VKRLYGPTDM***GG***)** and analyzed by ELISA using a panel of rat infect sera. Based on a receiver operating characteristics (ROC) curve (Figure 6B)**,** the area under the curve (AUC) for PsAg2-MAP4 varied from 0.9817 to 0.9978 (*p* ≥ 0.0001) as detected by ELISA with an interval confidence of 95%, demonstrating high diagnostic accuracy for the PsAg2 epitope (Figure 6).

## 4. Discussion

Our group has been working for some years on the characterization of GSC proteins from lower eukaryotes [53,54] and we observed that this group of proteins has a very low homology (~20–30%) to each other, a fact that has made their identification difficult in genomics and proteomics [55,56] projects. However, on the other hand, this information points out that these proteins are more likely to have specific epitopes under the immunological point of view.

Therefore, to overcome the difficulty of the diagnostic specificity of angiostrongyliasis, we characterize a protein from *A. cantonensis* described in the database as a putative presenilin protein using bioinformatics, biochemical and immunological tools. This first characterization of an atypical transmembrane aspartyl protease from *A. cantonensis*.

Intramembrane proteases catalyze the unusual cleavage of peptide bonds in the plane of biological membranes. They are categorized according to their active site. For example, the MEROPS database (MEROPS database. Available online: https://www.ebi.ac.uk/merops/ accessed on 18 November 2021) peptidase family, A22, contains membrane-inserted endopeptidases and aspartates residues in the active area are characteristics of peptidases of family A22. The GxGD aspartyl proteases comprise the classical human PS1 and PS2, the signal peptide peptidase (SPP), and SPP-like (SPPL) proteases [57,58] and five homologous presenilin proteins,

Presenilin is the catalytic portion of the gamma–secretase complex (GSC) along with other proteins, Anterior pharynx defective 1 (Aph1), Nicastrin, and presenilin enhancer 2 (Pen2), involved in complex stabilization and substrate binding. Most of the presenilins characterized until now are composed of 9-pass TM domains that undergo an endoproteolytic cleavage in the intracellular loop between TM6 and TM7, producing N-terminal and C-terminal fragments [59,60].

The predicted model for Ag-PS protein also presented 9 TM domains. Its topology is corroborated by the experiments of epitope identification (outside and inside) and contemplates the catalytic domain entailed of the conserved aspartates motifs XD (aa 241) and GLGD (aa 332-335), between TM6 and TM7, as occur in the human orthologue (YD; 192–194 and GLGD; 287–290) protein (Figure 2)**.** The presence of these critical motifs was also confirmed to be present in other worms such as *Caenorhabditis elegans* (P52166), *Angiostrongylus costaricensis* (A0A0R3PEK8), and *Schistosoma mansoni* (C0SKM6) (Figure 2).

However, the AgPS structure does not present other significant functional motifs such as the [GxxxG (GVKLG), SxxxGxxxxA (SVLVGKASA)] and the PALP. The first two motifs should be located at TMD8 of other eukaryotic proteins closely to the catalytic Asp and contribute to the helix packing. The PALP in the TMD9 stabilizes the GSC and characterizes the A22 family. This function has been demonstrated in studies performing mutations in the first Proline of PALP with leucine (P414L) which abrogated PS activity in *Drosophila melanogaster* and *C. elegans* [61].

Previously, a 46 kDa soluble aspartyl protease (GenBank accession no. NP 872129) from *A. cantonensis* was cloned and expressed and present in different infective levels larvae, young adults, and adult worms. Its primary function has been associated with exogenous hemoglobin degradation [62].

In this work, we describe, for the first time, the presence of an atypical presenilin-like aspartyl protease (AgPSat) in *A. cantonensis*. This nomenclature was attributed to proteins from different organisms that have high similarity with PSs but do not present certain domains characteristic of the PS family [62]. These non-catalytic activities may represent an ancestral PS function, described in non-mammalian animals, plants, and less complex eukaryotic organisms [63].

This sub-branch also possesses two conserved aspartic acid residues within adjacent predicted TM segments. However, the branch of presenilin homologous (PSH) proteins retains the conserved first Proline and the PALP motif but lacks several other motifs present in the A22 family [64,65]. Contrasting in the Ag-PS, the PALP motif is absent, but the protein has a proline located in the C-terminal fraction (aa 373). Interestingly, phylogenetic close related worms such as *Haemonchus contortus* (Figure 4) have the PALP motif. These collective data suggested that AgPS is an atypical aspartyl protease, with catalytic and non-catalytic functions essential for worm homeostasis [66,67,68,69,70,71].

The tissue and cellular location of this critical proteolytic enzyme and its cellular function need to be demonstrated.

To characterize this TM protein immunologically, a microarray of peptides was initially synthesized using the Spot-synthesis technique to identify linear B-cell epitopes in the putative PS of *A. cantonesis* that are recognized by infected rats IgG antibodies. Twelve epitopes were readily defined (Table 1), confirming the predicted AgPSat protein’s likelihood to be expressed. In addition, the distribution of the peptides corroborates the topological model proposed for the AgPSat.

All peptides sequences were apparently specifically present in the *A. cantonensis* protein from the determinant profiles. Thus, initially, the immunogenicity of the 12 peptides was demonstrated by an ELISA-peptide assay using a pool of infected rats sera (Figure 4). An ELISA assay was developed using one of the peptides that highly showed sensitivity and specificity of MAP4, displaying its potential role to overcome the limitations of *A. cantonensis* diagnostics.

## 5. Conclusions

In this work, we have demonstrated that the putative aspartyl protease C7BVX5 (UNIPROT) is a 9-pass transmembrane atypical presenilin and proposed a structural model of membrane insertion. In addition, we have provided new insights describing, with more excellent coverage, the set of 12 IgG linear B-epitopes for this aspartyl protease from *A. cantonensis* and demonstrated the specificity and eligibility of some epitopes to enter phase IIB studies to develop new and fast diagnostic assays for angiostrongyliasis.

## Figures and Tables

**Figure 1 membranes-12-00108-f001:**
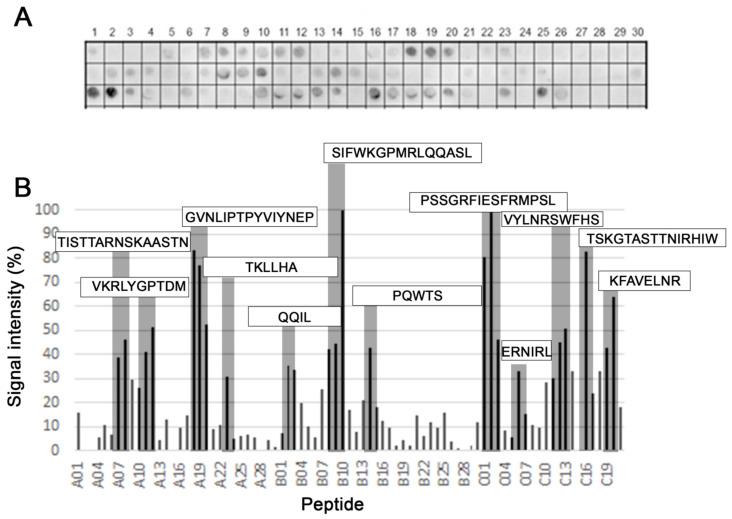
B-cell linear epitopes mapping along the primary sequence (415 aa, 81 peptides) of the aspartyl protease from *A. cantonensis* in the SPOT synthesis array (15 residues with overlapping of 10) with the sera (*n* = 5) of infected rats. (**A**) IgG-reactive peptides; (**B**) signal intensity (SI). The peptides represented the coding region of PS protein are listed in Appendix A. Each peptide was identified by the Spot-synthesis membrane position numbering. Spot intensities below 20% were considered as background. The positive peptides identified and assessed for overlap are gray and the resulting sequence within the rectangles.

**Figure 2 membranes-12-00108-f002:**
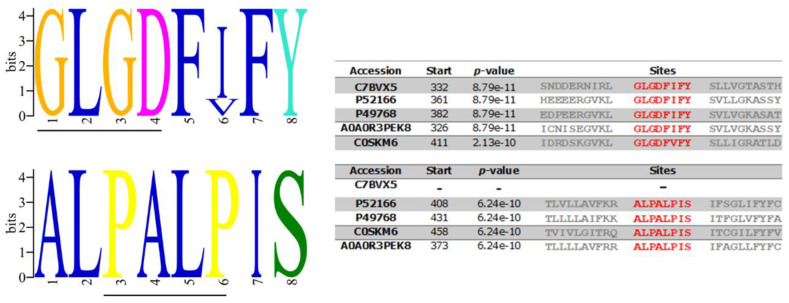
Sequence logo of *Angiostrongylus cantonensis* (C7BVX5) presenilin-like protein conserved motifs. Multiple sequence alignment of different organisms: *Caenorhabditis elegans* (P52166), Homo sapiens PS1 (P49768), *Angiostrongylus costaricensis* (A0A0R3PEK8), and *Schistosoma mansoni* (C0SKM6), show amino acid variations in significant PS motifs GLGD and PALP.

**Figure 3 membranes-12-00108-f003:**
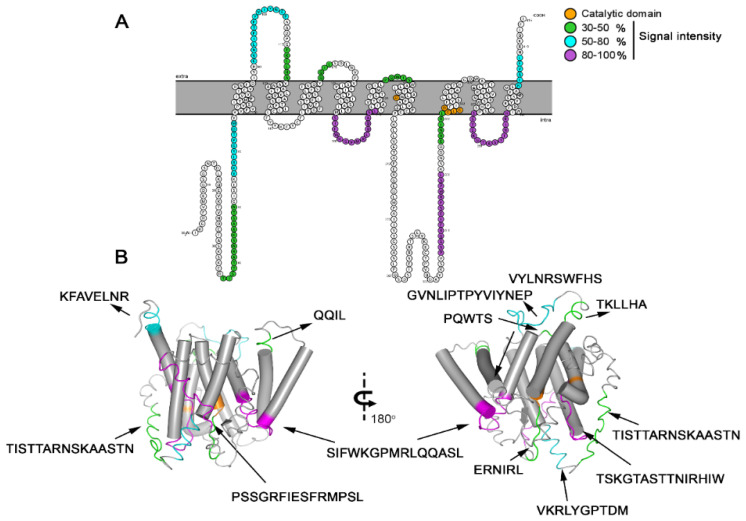
Model of Ag/PS as a multi-pass transmembrane protein (TM), based on the predictive results obtained by bioinformatics (available online: https://topcons.net/pred accessed on 12 November 2020) and layout generated using Protter (available online: http://wlab.ethz.ch/protter/ accessed on 15 November 2020). The 9 TM domains reveal 12 identified epitopes and the catalytic pocket (orange) with the typical motifs XD (TMD6) and GLGD (TMD7) but the absence of the PALP domain (**A**). Three-dimensional structure and membrane-spanning model of *A. cantonensis* presenilin protein. 3D protein conformation and structure were determined using the I-Tasser, and epitopes were highlighted (purple), indicating the position of the 12 linear B epitopes identified by Spot-synthesis ((**B**), Table 1).

**Figure 4 membranes-12-00108-f004:**
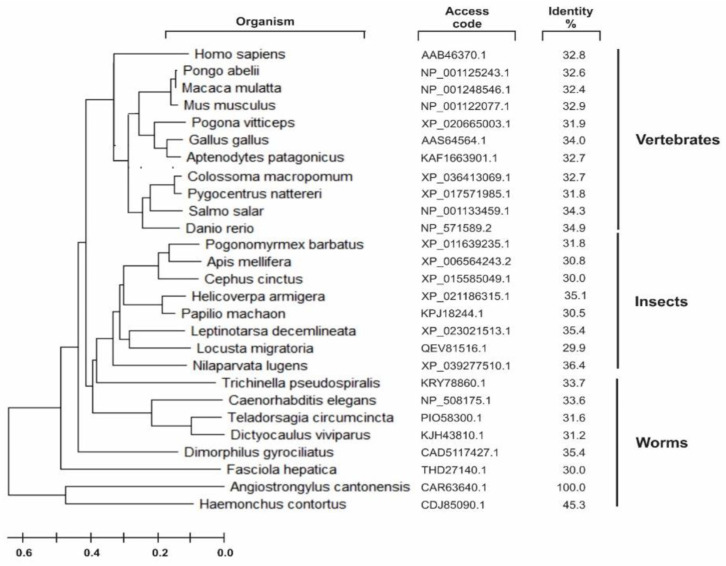
Phylogenetic relationship of PS-like proteins and protein–protein interaction network of *Angiostrongylus cantonensis* putative PS-like (Uniprot C7BVX5_ANGCA) aspartyl protease. Amino acid sequences obtained from Uniprot server (http://www.uniprot.org accessed on 15 December 2021, search criteria name: “PS” and length: [300 to 800]) were aligned with Clustal W, and the phylogenetic tree was constructed with the sequences aligned by the neighbor-joining algorithm using a CLUSTAL W in MEGA software. Family members are grouped according to their relationship to human SPP/SPPL orthologues.

**Figure 5 membranes-12-00108-f005:**
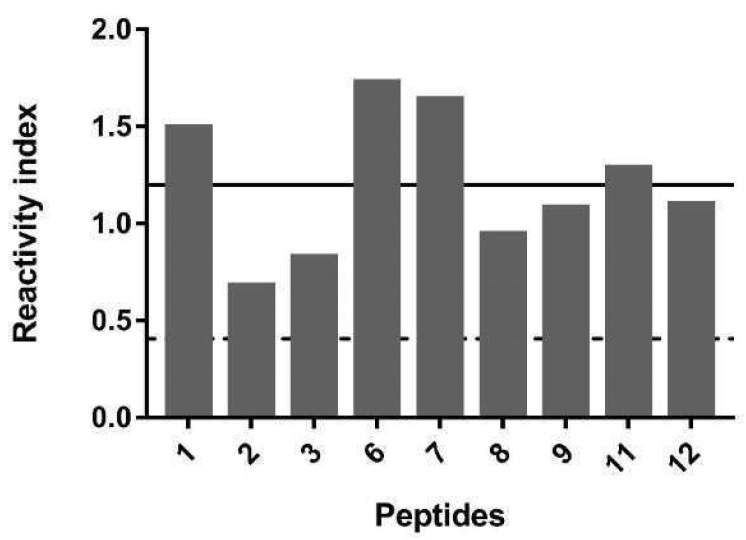
Reactivity of nine IgG peptides (PSAg1-PSAg3, PSAg6-PSAg9, PSAg11, and PSAg12; Table 1) with pool (*n* = 16) of serum from infected rat mice by IgG-ELISA-peptide. The solid and broken lines indicate the mean values of reactivity found with the set of peptides analyzed with positive sera (sera from infected rats) and negative sera (sera from healthy rats), respectively. Columns show the average of duplicates.

**Figure 6 membranes-12-00108-f006:**
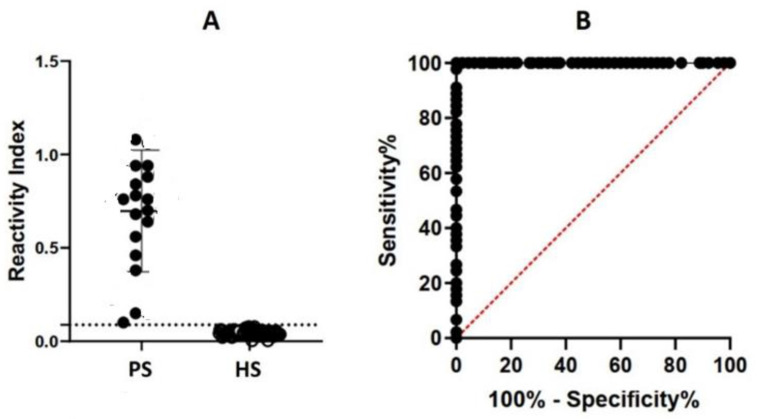
Reactivity of rat infected sera (PS, *n* = 16) and healthy rat sera (HS, *n* = 25) with synthetic specific spanning epitope PsAg2-MAP4 in an ELISA. (**A**) The y-axis shows sera’s mean reactivity (positive/mean negative) from infected rats. The ROC analysis (**B**) using PS showed that the sensitivity and specificity of the in-house ELISA-peptide was 100% for PsAg2. The dashed line shows the cut-off of the assay.

**Table 1 membranes-12-00108-t001:** List of the identified PsAg ratIgG epitopes and structural properties.

Code	Epitope	Position aa Number	Secondary Structure ^1^	Cross-Reactivity Sequence ^2^ (UniProtKB)	Soluble Synthesized Peptide Sequences ^3^
**PsAg1**	**TISTTARNSKAASTN**	**36–50**	**C**	NQP	**TISTTARNSKAASTN**
**PsAg2**	**VKRLYGPTDM**	**56–65**	**C**	**NQP**	**GSGVKRLYGPTDMGG**
**PsAg3**	** *GVNLI* ** **PTPY*VIYNEP***	**91–105**	**H + C + H**	**NQP**	**GVNLIPTPYVIYNEP**
**PsAg4**	** *TKLLHA* **	**111–116**	**H**	**NQP**	**TKLLHAVANAATFLV**
**PsAg5**	** *QQIL* **	**167–170**	**H**	**NQP**	**GSGFSFVQYQQILGG**
**PsAg6**	**SIFWKGP*MRLQQASL***	**196–210**	**C + H**	**NQP**	**SIFWKGPMRLQQASL**
**PsAg7**	**PQ*WTS***	**216–230**	**C + H**	**NQP**	**TVTLTIMQILPQWTS**
**PsAg8**	**PSSGRFIESFRMPSL**	**306–320**	**C**	**NQP**	**PSSGRFIESFRMPSL**
**PsAg9**	**ERNIRLGLG**	**326–334**	**C**	**NQP**	**ERNIRLGLGDFIFYS**
**PsAg10**	** *VYLNRSWFHS* **	**359–366**	**H**	**NQP**	**GSGVYLNRSWFHSGG**
**PsAg11**	** *TSKGTA* ** **STTN*IRHIW***	**374–388**	**H + C + H**	**NQP**	**TSKGTASTTNIRHIW**
**PsAg12**	**KFAVELNR**	**401–408**	**C**	**NQP**	**SSRFAMSKFAVELNR**

^1^ Predicted by I-TASSER (accessed on 10 March 2021); ^2^ Peptide search database UniProtKB; http://www.uniprot.org/ (accessed on 20 March 2021); NQP, no query peptide (Pir Protein, accessed on 7 April 2021); ^3^ The amino acids marked in the synthetic peptides are not part of the epitope but were inserted to complement 15 mer. C, coil; H, helix.

## Data Availability

The data presented in this study are available on request from the corresponding author.

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
