# Peer review of "Angiostrongylus cantonensis an Atypical Presenilin: Epitope Mapping, Characterization, and Development of an ELISA Peptide Assay for Specific Diagnostic of Angiostrongyliasis"

_membranes, 2022, doi:10.3390/membranes12020108_

Round 1

Reviewer 1 Report

Peptides of aspartyl protease from Angiostrongilus cantonensis, which causes Angiostrongyliasis, for antibody test for diagnosis were determined using a peptide microarray. It is unclear why aspartyl protease was targeted for the analysis, because various proteins are suggested as unique antigen for detection of A. cantonensis recently. The authors only mention aspartyl protease from Trypanosoma cruzi in the bottom of introduction (Antigens from distinct nematodes should differ). The authors should explain the reason definitely.

Author Response

Referee 1

Peptides of aspartyl protease from Angiostrongilus cantonensis, which causes Angiostrongyliasis, were determined for antibody test for diagnosis using a peptide microarray. 

1)It is unclear why target aspartyl protease for the analysis is because various proteins have recently been suggested as unique antigens for detecting A. cantonensis. However, the authors only mention aspartyl protease from Trypanosoma cruzi at the bottom of the introduction (Antigens from distinct nematodes should differ). The authors should explain the reason.

R: We are grateful for the referee's question, and we agree that the actual text was confusing. We apologize for having sent the wrong version of the manuscript.

The reasons for using aspartyl protease as a target diagnosis for angiostrongyliasis are clearly explained in this review.

The reasons A general review of the introduction and discussion is valid, adapting to these questions. We could not find a reference that had used purified Angiostrongilus antigen but did not develop diagnostic immunological tests for Angiostrongiliase. Or the only work wrapping this issue was using a purified cloned recombinant protein. In the meantime, none of these works investigated a possible cross-reaction of two antigens using sera from patients with other pathologies, including this work that used a cloned protein. The modifications introduced are marked in yellow.

Reviewer 2 Report

In the following paper, entitled “Angiostrongilus cantonensis aspartyl protease: Epitope mapping, characterization, and development of an ELISA peptide assay for specific diagnostic of angiostrongyliasis”, authors identified several B-linear epitopes by using peptide microarray approach, from which two epitopes with higher cross-reactivity with A. costaricensis. I have some major concerns, which are below:
1. It is not clear why authors only considered aspartyl protease for identifying B-linear epitopes. Since a number of proteins are available (as they mentioned in the introduction), it is possible that there could be more potent antigenic proteins that share antigenic regions that could induce higher sensitivity than the current one. 
2. In methods, the authors mentioned protein-protein interaction analysis; however, I do not see the result in figure 4 or the result and discussion. 
3. In section 3.3, "Potential homologs were identified by PSI-BLAST and the level of similarity among the PSs from these species was aligned to the Homo sapiens sequence (data not shown)" , please include this data as supplementary.
4. Authors should consider their article for English editing. The arrangement and structure of the article are not well managed for the general reader. They mixed the Presenilin/PS/ aspartyl protease terminology in the whole manuscript. Authors should discuss the Presenilin protein in the background of the study, why it was only chosen, why not about others? 

Author Response

Referee 2

The following paper, entitled “Angiostrongilus cantonensis aspartyl protease: Epitope mapping, characterization, and development of an ELISA peptide assay for specific diagnostic of angiostrongyliasis,” authors identified several B-linear epitopes by using peptide microarray approach, from which two epitopes with higher cross-reactivity with A. costaricensis. I have some significant concerns, which are below:

  1. It is unclear why authors only considered aspartyl protease to identify B-linear epitopes. Since several proteins are available (as they mentioned in the introduction), it is possible that there could be more potent antigenic proteins that share antigenic regions that could induce higher sensitivity than the current one. 

R: Thousands of proteins can be used for mapping. Meanwhile, mapping is expensive and should be undertaken rationally. The reasons that led us to choose the presenilin group of aspartyl protease are now explicitly in the manuscript. Furthermore, years of experience expended working with this group of proteins in different microorganisms. Therefore, they present low similarity and a high probability of finding specific epitopes.

  1. In methods, the authors mentioned protein-protein interaction analysis; however, I do not see the result in figure 4 or the development and discussion.

R: These words were removed; please see line 201.

  1. In section 3.3, "Potential homologs were identified by PSI-BLAST and the level of similarity among the PSs from these species was aligned to the Homo sapiens sequence (data not shown)," please include this data as supplementary.

R: It was necessary to modify this paragraph, once searching the definition of "homologous" used in other proteins, we concluded that the AgPS, does not fit into this group due to the absence of the PALP domain. Therefore, we rename the protein as "Atypical presenilin," including in the title of work. Examples in literature were cited, not text. A new supplementary Figure 5 was annexed comparing the possible function of the protein.

  1. Authors should consider their articles for English editing. The arrangement and structure of the paper are not well managed for the general reader. They mixed the Presenilin / PS/aspartyl protease terminology in the whole manuscript. Authors should discuss the Presenilin protein in the background of the study, why it was only chosen, why not about others?

R: We apologize that we have not sent a final version of the manuscript. Now the manuscript was updated and substantial modifications, mainly in the introduction and discussion, were introduced. Therefore, we certify that using the English editing resources is not necessary. The manuscript was also reviewed by a native researcher the Dr. David W. Provance.

The new version extensively discussed the reasons that led us to use presenilin as an antigen. In addition, we try to standardize the possible Presenilin / PS/aspartyl protease therms. Meanwhile, each word has a different meaning. The modifications introduced are marked in yellow.

Round 2

Reviewer 2 Report

The authors addressed all of my comments, and I recommend this paper for acceptance.